# Xrp1 and Irbp18 trigger a feed-forward loop of proteotoxic stress to induce the loser status

**Paul F. Langton**⊙*, **Michael E. Baumgartner**⊙¤, **Remi Logeay, Eugenia Piddini**⊙*

School of Cellular and Molecular Medicine, University of Bristol, Bristol, United Kingdom

¤ Current address: Perelman School of Medicine, University of Pennsylvania, Philadelphia, Pennsylvania, United States of America

* paul.langton@bristol.ac.uk (PFL); eugenia.piddini@bristol.ac.uk (EP)

**Data Availability Statement:** All relevant data are within the manuscript and its Supporting Information files.

**Funding:** This work was supported by a Wellcome Trust Senior Research Fellowship to E.P. (205010/

## Abstract

Cell competition induces the elimination of less-fit "loser" cells by fitter "winner" cells. In *Drosophila*, cells heterozygous mutant in ribosome genes, *Rp/+*, known as *Minutes*, are outcompeted by wild-type cells. *Rp/+* cells display proteotoxic stress and the oxidative stress response, which drive the loser status. Minute cell competition also requires the transcription factors Irbp18 and Xrp1, but how these contribute to the loser status is partially understood. Here we provide evidence that initial proteotoxic stress in *RpS3/+* cells is Xrp1-independent. However, Xrp1 is sufficient to induce proteotoxic stress in otherwise wild-type cells and is necessary for the high levels of proteotoxic stress found in *RpS3/+* cells. Surprisingly, Xrp1 is also induced downstream of proteotoxic stress, and is required for the competitive elimination of cells suffering from proteotoxic stress or overexpressing Nrf2. Our data suggests that a feed-forward loop between Xrp1, proteotoxic stress, and Nrf2 drives Minute cells to become losers.

## Author summary

Removal of damaged cells is important for maintaining tissue health and preventing disease. Cells that become damaged by mutation or due to aging are actively eliminated from tissues by their fitter neighbouring cells through a process called cell competition. Cell competition was discovered in *Drosophila* through the study of Minute mutants, which are a class of mutations in ribosomal genes. Cells carrying a mutation in Minute genes are said to behave as losers, as over time, they are eliminated by competition with surrounding wild type cells. It is known that toxic protein aggregates in the cytoplasm contribute to the loser status of Minute cells. The factors Xrp1 and Irbp18 are also required for the elimination of Minute cells. Here we uncover the relationship between these factors and toxic protein aggregates in cell competition. We find that Xrp1 and Irbp18 promote protein aggregate formation and that, vice versa, protein aggregates induce Xrp1 and Irbp18 activity. This amplifies stress signalling and reduces the fitness of Minute cells, leading to their removal from tissues through cell competition. These findings shed light on an important

Z/, 16/Z) and a Cancer Research UK Programme Foundation Award to E.P. (Grant C38607/A26831). The funders had no role in study design, data collection and analysis, decision to publish, or preparation of the manuscript.

**Competing interests:** The authors have declared that no competing interests exist.

mechanism by which cells carrying certain types of damage can be eliminated to preserve organism health.

## Introduction

Cells within a tissue may become damaged due to spontaneous or environmentally induced mutations, and it is beneficial to organismal health if these cells are removed and replaced by healthy cells. During cell competition, fitter cells, termed winners, recognise and eliminate less-fit cells, termed losers, resulting in restoration of tissue homoeostasis [1–3]. Cell competition therefore promotes tissue health and is thought to provide a level of protection against developmental aberrations [4–6] and against cancer by removing cells carrying oncoplastic mutations [1,7]. However, an increasing body of evidence indicates that cell competition can also promote growth of established tumours, enabling them to expand at the expense of surrounding healthy cells [7,8].

Minute cell competition was discovered through the study of a class of *Drosophila* ribosomal mutations called *Minutes* [9] and initial work suggests that it is conserved in mammals [10]. While homozygous *Rp* mutations are mostly cell lethal, heterozygosity for most *Rp* mutations gives rise to viable adult flies that exhibit a range of phenotypes including developmental delay and shortened macrochaete bristles [9,11]. *Rp/+* tissues display a higher cell-autonomous death frequency than wild-type tissues [12–15], and competitive interactions further elevate cell death in *Rp/+* cells bordering wild-type cells, contributing to progressive loss of *Rp/+* cells over time [14,16,17].

It was suggested that *Rp/+* cells are eliminated by cell competition due to their reduced translation rate [3,18–22]. However, we and others have recently shown that *Rp/+* cells experience significant proteotoxic stress and this is the main driver of their loser status [13,14]. *Rp/+* cells have a stoichiometric imbalance of ribosome subunits, which may provide the source of proteotoxic stress. The autophagy and proteasomal machineries become overloaded and protein aggregates build up in *Rp/+* cells, leading to activation of stress pathways. This includes activation of Nuclear factor erythroid 2-related factor 2 (Nrf2) and of the oxidative stress response [23], which we have shown to be sufficient to cause the loser status [24]. Restoring proteostasis in *Rp/+* cells suppresses the activation of the oxidative stress response and inhibits both autonomous and competitive cell death [13,14].

Genetic screening for suppressors of cell competition led to the identification of Xrp1 [20,25,26], a basic leucine Zipper (bZip) transcription factor. Loss of Xrp1 rescues both the reduced growth and competitive cell death of *Rp/+* cells in mosaic tissues [20,25]. Consistently, loss of Xrp1 restores translation rates and abolishes the increased JNK pathway activity characteristic of *Rp/+* cells [20]. Xrp1 forms heterodimers with another bZip transcription factor called Inverted repeat binding protein 18kDa (Irbp18) [27,28], and removal of Irbp18 also strongly suppresses the competitive elimination of *Rp/+* cells in mosaic tissues [29]. Irbp18 and Xrp1 are transcriptionally upregulated and mutually required for each other's expression in *Rp/+* cells, suggesting they function together in Minute cell competition [29]. Irbp18 forms heterodimers with another bZip transcription factor, ATF4 [28]. However, knockdown of ATF4 in *Rp/+* cells reduces their survival in mosaic tissues, which is the opposite effect to knockdown of Xrp1 or Irbp18. This has been interpreted to suggest that the ATF4-Irbp18 heterodimer acts independently to the Xrp1-Irbp18 heterodimer [29].

How the Xrp1/Irbp18 complex contributes to the loser status is not clear. Given the recently identified role of proteotoxic stress in cell competition we sought to establish whether Xrp1/

Irbp18 and proteotoxic stress act independently or in the same pathway to contribute to cell competition in *Rp/+* cells. We identify a feed-forward loop between Xrp1/Irbp18 and proteotoxic stress, which is required for downstream activation of the oxidative stress response and the loser status. Our data suggests a model in which the initial insult in *RpS3/+* cells is ribosomal imbalance-induced proteotoxic stress, which is Xrp1 independent. Xrp1 is then transcriptionally activated downstream of proteotoxic stress, by increased phosphorylated-eukaryotic Initiation Factor 2α (p-eIF2α), and possibly by Nrf2. The Xrp1-Irbp18 complex then induces further proteotoxic stress, completing the feed-forward loop. This work provides new insight into the interactions between the stress signalling pathways active in *Rp/+* cells and provides a mechanism for how the Xrp1-Irbp18 heterodimer mediates the competitive elimination of *Rp/+* cells by wild-type cells.

## Results

To probe the role of the Xrp1-Irbp18 complex in *Rp/+* cells, we first established whether RNAi lines against each functionally knock-down these genes. *Xrp1* expression depends on its own activity [25,29] and on the activity of Irbp18 [29]. As expected, knockdown of Xrp1 (*xrp1^{KK104477}* RNAi line, hereafter referred to as *xrp1-RNAi*) in the posterior compartment of wild type wing discs reduced expression of an *xrp1* transcriptional reporter, *xrp1-lacZ* (S1A and S1B Fig). Similarly, knockdown of Irbp18 (*irbp18^{KK110056}* RNAi line, hereafter referred to as *irbp18-RNAi*) reduced levels of *xrp1-lacZ* (S1C and S1D Fig). Mutations in *xrp1* and *irbp18* prevent *Rp/+* cells from being out-competed by wild-type cells in mosaic tissues [20,25,26,29]. Accordingly, knockdown of Xrp1 or Irbp18 rescued the competitive elimination of *RpS3/+* cells in wing discs. Compared to *RpS3/+* patches, *RpS3/+* patches expressing *xrp1-RNAi* (S1E–S1G Fig), or *irbp18-RNAi* (S1H–S1J Fig) grew substantially larger. These data indicate that those RNAi lines effectively knockdown Xrp1 and Irbp18.

To investigate the role of Xrp1 and Irbp18 in proteotoxic stress and the oxidative stress response, which are primary drivers of the loser status in *Rp/+* cells [13,14,24], we expressed *xrp1-RNAi* specifically in the posterior compartment of *RpS3/+* wing discs with the *hedgehog (hh)-gal4* driver. Xrp1 knockdown significantly rescued the accumulation of p-eIF2α (Fig 1A and 1B), a marker of the integrated stress response, which is induced in response to proteotoxic stress [30,31] and is upregulated in *RpS3/+* cells [13,14]. Xrp1 knockdown also strongly inhibited the oxidative stress response in *RpS3/+* cells, as it reduced the expression of Glutathione S transferase D1-GFP (GstD1-GFP) (Fig 1A and 1C), a reporter of Nrf2 [32]. Irbp18 knockdown also rescued both p-eIF2α upregulation and GstD1-GFP upregulation in *RpS3/+* discs (Fig 1D–1F). Refractory to sigma P (Ref(2)p), also known as p62, is an autophagy adaptor and cargo [33] and a marker of cytosolic protein aggregates [34], which accumulates in *RpS3/+* cells due to proteotoxic stress overload [14]. The accumulation of p62-labelled aggregates in *RpS3/+* cells was rescued both by *xrp1-RNAi* (Fig 1G and 1H) and by *irbp18-RNAi* (Fig 1I and 1J), further indicating that proteotoxic stress in *Rp/+* cells is mediated by the Xrp1/Irbp18 complex. Together, these data show that Xrp1 and Irbp18 are required for, and act upstream of, proteotoxic stress and the oxidative stress response in *RpS3/+* cells.

*Rp/+* cells have recently been shown to have a stoichiometric imbalance in their ribosome subunits, suggesting that this is the initial proteostatic perturbation leading to proteotoxic stress. Specifically, *Rp/+* cells have an excess of large-subunit (LSU) proteins and a reduced complement of small-subunit (SSU) proteins, relative to wild-type cells [13,14]. The data in Fig 1B, 1C, 1E, 1F, 1H and 1J indicate that Xrp1 and Irbp18 induce proteotoxic stress in *RpS3/+* cells, therefore we asked whether the ribosomal imbalance in *RpS3/+* cells is also downstream of Xrp1. Proteomic analysis revealed that removal of one copy of *xrp1*, which is

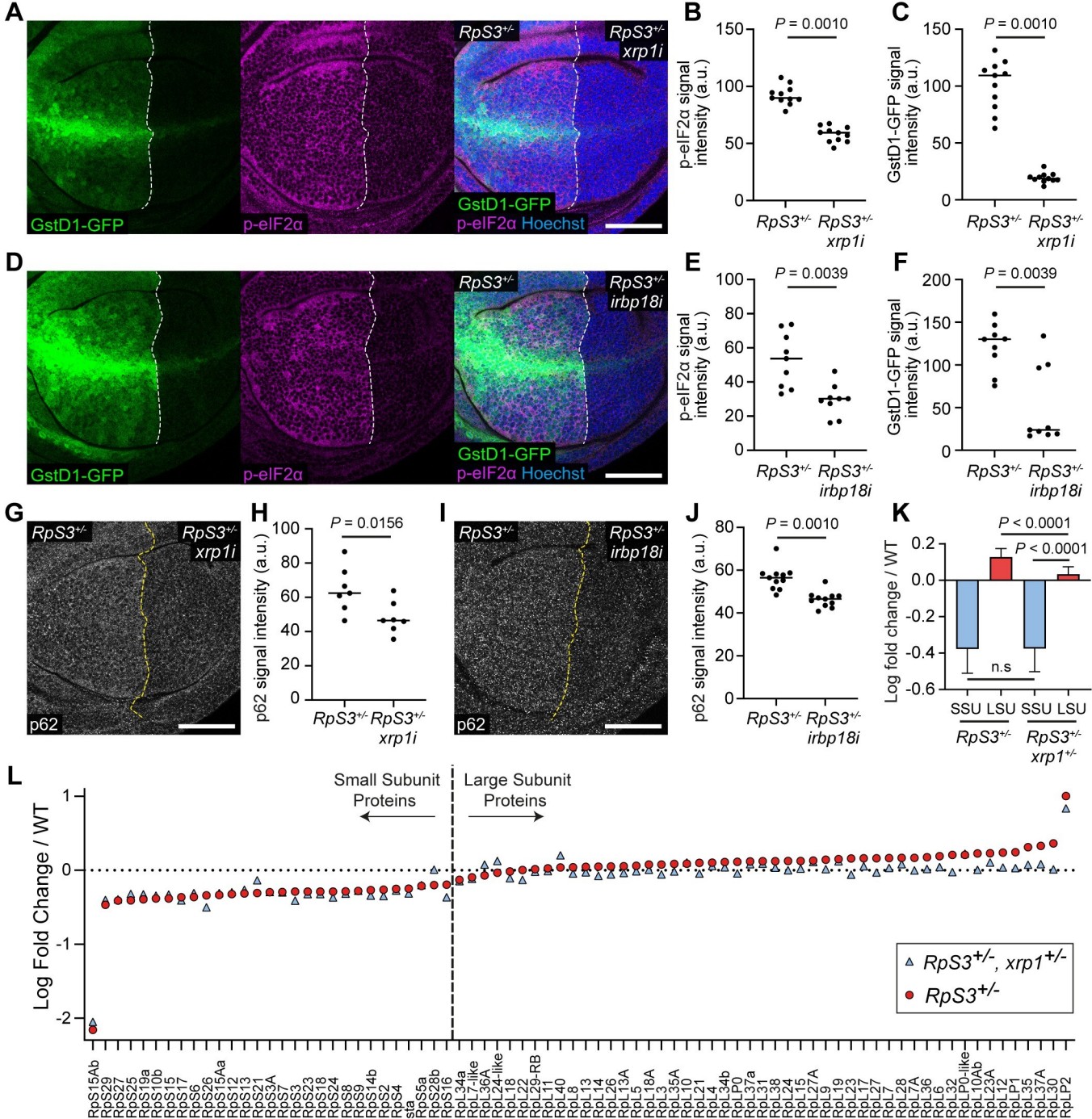

**Fig 1. Xrp1 and Irbp18 are required for proteotoxic stress and the oxidative stress response induced by Rp loss.** (**A-C**) An *RpS3*$^{+/-}$ wing disc harboring the GstD1-GFP reporter (green) and expressing *xrp1-RNAi* (*xrp1i*) in the posterior compartment, immuno-stained for p-eIF2α (magenta) with nuclei labelled in blue (**A**). Quantifications of p-eIF2α signal intensity (n = 11; two-sided Wilcoxon signed-rank test) and GstD1-GFP signal intensity (n = 11; two-sided Wilcoxon signed-rank test) are shown in (**B**) and (**C**) respectively. (**D-F**) An *RpS3*$^{+/-}$ wing disc harboring the GstD1-GFP reporter (green) and expressing *irbp18-RNAi* (*irbp18i*) in the posterior compartment, immuno-stained for p-eIF2α (magenta) with nuclei labelled in blue (**D**). Quantifications of p-eIF2α signal intensity (n = 9; two-sided Wilcoxon signed-rank test) and GstD1-GFP signal intensity (n = 9; two-sided Wilcoxon signed-rank test) are shown in (**E**) and (**F**) respectively. (**G-H**) A wing disc of the same genotype as shown in (**A**), immuno-stained for p62 (grey) (**G**), with quantification of p62 signal intensity (**H**) (n = 7; two-sided Wilcoxon signed-rank test). (**I-J**) A wing disc of the same genotype as shown in (**D**), immuno-stained for p62 (grey) (**I**), with quantification of p62 signal intensity (**J**) (n = 11; two-sided Wilcoxon signed-rank test). (**K**) A bar graph showing the mean log fold change in all Small-subunit (SSU) and Large-subunit (LSU) ribosomal proteins detected by mass spectrometry in *RpS3*$^{+/-}$ and *RpS3*$^{+/-}$, *Xrp1*$^{+/-}$ wing discs relative to wild-type discs, as indicated (n = 29; two-sided Wilcoxon signed-rank test for comparison of SSU, n = 49; two-sided Wilcoxon signed-rank test for comparison of LSU, n = 29 and 49, respectively;

two-sided Mann–Whitney U-test for comparison of SSU and LSU in $RpS3^{+/-}$, $Xrp1^{+/-}$ wing discs), error bars represent 95% confidence interval. (**L**) Mean log fold change in SSU and LSU ribosomal proteins detected by mass spectrometry (n = 2) in $RpS3^{+/-}$ and $RpS3^{+/-}$, $Xrp1^{+/-}$ wing discs relative to wild-type discs, as indicated. In this figure and throughout: scale bars are 50μm; dashed white or yellow lines mark compartment boundaries; each data point on the scatter plots represents one wing disc or one wing disc compartment and the horizontal line represents the median; all n values refer to the number of individual wing discs except for Fig 1K and 1L; posterior is right and dorsal is up.

sufficient to rescue $Rp/+$ cells from competition [20], rescues the excess of LSU proteins but does not affect the reduction in SSU proteins (Fig 1K and 1L). Thus, SSU protein imbalance in $RpS3/+$ cells is independent of Xrp1. This suggests that the initial proteotoxic stress experienced by $Rp/+$ cells is an SSU/LSU stoichiometric imbalance. This may provide the signal for Xrp1 induction, which in turn exacerbates proteotoxic stress, resulting in accumulation of LSU proteins.

Mutations in the E3 ubiquitin ligase encoding gene *mahjong* (*mahj)* lead to the loser status, and $mahj^{-/-}$ cells are out-competed by wild-type cells in mosaic tissues [35]. Although Mahj is functionally distinct to ribosomal proteins, the gene expression signatures of *mahj* and *RpS3* mutants significantly overlap, indicating a common mechanism leading to the loser status [24]. Indeed, *mahj* cells also show upregulation of p62 labelled aggregates suggesting that they experience proteotoxic stress [14]. Interestingly, Xrp1 knockdown rescued *mahj*-RNAi expressing cells from elimination in mosaic wing discs (Fig 2A–2C). We then induced larger patches of *mahj-RNAi* cells and found that, like $Rp/+$ cells, *mahj-RNAi* expressing cells upregulate p-eIF2α (Fig 2D). Simultaneous Xrp1 knockdown rescued the accumulation of p-eIF2α (Fig 2E and 2F), suggesting that Xrp1 is also upstream of proteotoxic stress in *mahj* deficient cells. The rescue was not due to the presence of a second UAS construct (*UAS-xrp1-RNAi)*, which could have weakened the expression of *UAS-mahj-RNAi* by titrating Gal4, as the cells expressing *mahj-RNAi* in fact also carried a second control UAS construct ($40D^{UAS}$), which does not affect readouts of the loser status (S2 Fig). All further experiments in this study that compare the phenotype of expression of a single UAS construct to that of two UAS constructs use this strategy. Thus, Xrp1 contributes to the competitive elimination of cells with distinct loser backgrounds, $Rp/+$ and *mahj*, which are both linked to proteotoxic stress.

The results described above suggest that Xrp1 functions upstream of proteotoxic stress and oxidative stress in $RpS3/+$ and *mahj* deficient cells, so we asked whether Xrp1 is sufficient to induce proteotoxic stress. We over-expressed the $xrp1^{long}$ isoform [36] in the posterior compartment of wing discs with the *engrailed (en)-gal4* driver and found this condition to be larval lethal before the 3$^{rd}$ instar, which is consistent with previous reports that *xrp1* over-expression induces expression of proapoptotic genes [25] and high levels of cell death [25,29,36,37]. To circumvent this lethality, we used a temperature sensitive Gal4 inhibitor, Gal80$^{ts}$, to prevent *xrp1* expression throughout most of larval development. Shifting the larvae to the Gal80$^{ts}$ restrictive temperature 24 hours before dissection allowed for a relatively short burst of *xrp1* expression. Under these conditions, *xrp1* over-expressing compartments accumulated GstD1-GFP (Fig 3A and 3B), p62 (Fig 3C and 3D) and had higher levels of p-eIF2α (Fig 3E and 3F) than the wild-type, control compartments. Therefore, Xrp1 is sufficient to induce proteotoxic stress. Conversely, Irbp18 overexpression did not increase p-eIF2α or p62 (S3A–S3D Fig) suggesting that Irbp18 alone is not sufficient to induce proteotoxic stress, which is compatible with the observation that Xrp1-, but not Irbp18-overexpressing cells, are eliminated from mosaic tissues [29]. Overexpression of an inert protein, GFP, also did not upregulate markers of proteotoxic stress (S3E–S3H Fig), confirming that the effects of Xrp1 are not due to overexpression *per se*. We next asked whether Irbp18 is required for Xrp1 to generate proteotoxic stress. Again, using Gal80$^{ts}$ to control transgene expression, we found that a 24h burst of *irbp18-RNAi* expression in wild-type discs was able to reduce *xrp1* expression (S4A Fig),

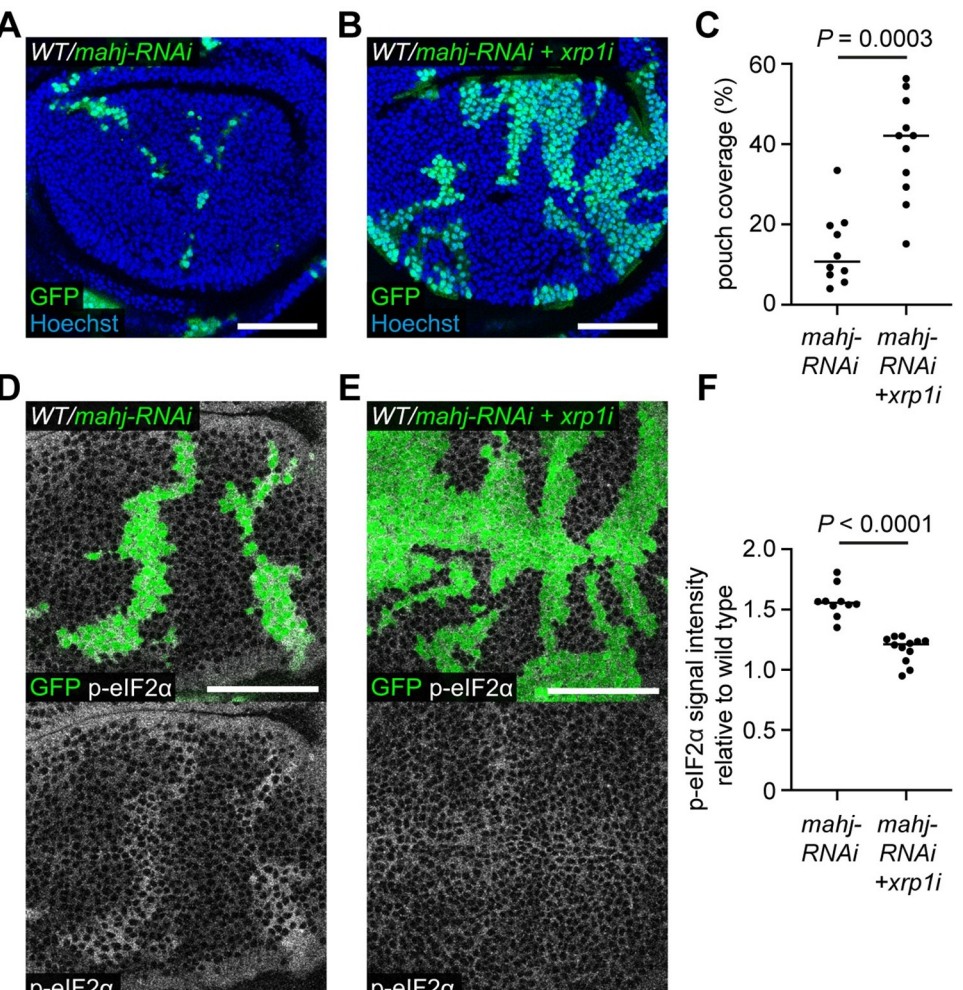

**Fig 2. Xrp1 is required for competitive elimination of *mahjong* mutant cells.** (**A-C**) Wild-type wing discs harboring *mahj-RNAi* cells (GFP positive) (**A**) or *mahj-RNAi* cells also expressing *xrp1-RNAi* (GFP positive) (**B**) with nuclei labelled in blue, and quantification of percentage coverage of the pouch (**C**) (n = 10 and 11, respectively; two-sided Mann–Whitney U-test). (**D-F**) Wild-type wing discs harboring *mahj-RNAi* cells (GFP positive) (**D**) or *mahj-RNAi* cells also expressing *xrp1-RNAi* (GFP positive) (**E**) immuno-stained for p-eIF2α (grey) with quantification of p-eIF2α signal intensity relative to wild-type cells (**F**) (n = 10 and 12, respectively; two-sided Mann–Whitney U-test).

suggesting that Irbp18 can be effectively knocked down over this time window. Under this regime, *xrp1* and *irbp18-RNAi* expressing cells had lower levels of p62 labelled aggregates than *xrp1* expressing cells (S4B–S4D Fig), suggesting that Irbp18 is, at least partially, required for Xrp1 to activate proteotoxic stress. This is consistent with the earlier observation that the elimination of Xrp1 overexpressing cells from mosaic tissues can be rescued by a mutation in *Irbp18* [29]. Overall, this data indicates that Xrp1, along with Irbp18, is responsible for inducing proteotoxic stress and the oxidative stress response in *Rp/+* cells, which may explain why removal of Xrp1 or Irbp18 so effectively rescues Minute competition.

If Xrp1 and Irbp18 are required in cell competition because they induce proteotoxic stress, then inducing proteotoxic stress by other means should lead to the loser status in an Xrp1- and Irbp18- independent manner. To test this hypothesis, we induced proteotoxic stress by well-established means. eIF2α is phosphorylated in response to proteotoxic stress, leading to global attenuation of translation [30,31]. However, sustained increase in p-eIF2α has also been

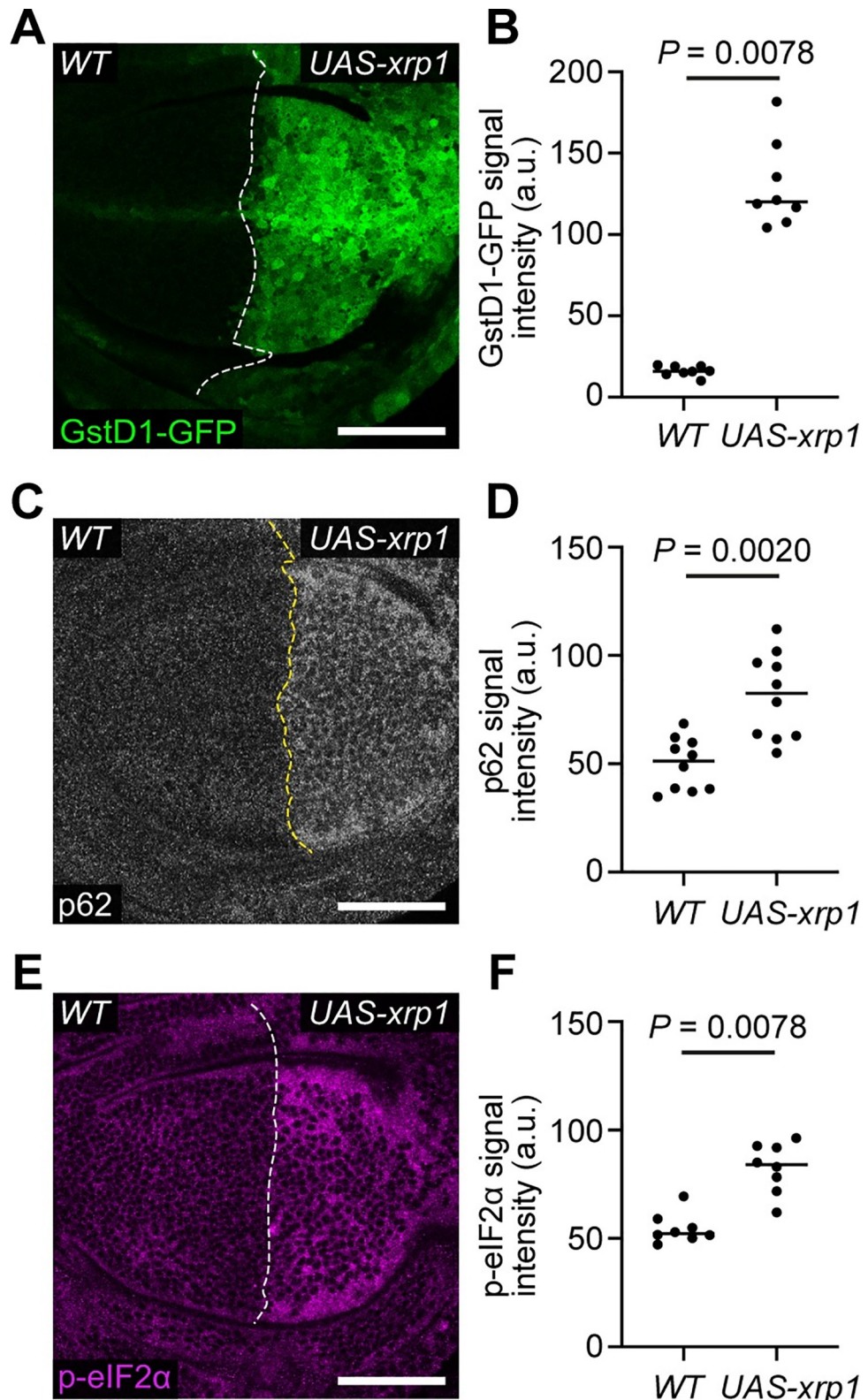

**Fig 3. Xrp1 is sufficient for proteotoxic stress and the oxidative stress response.** (**A-B**) A wild-type (WT) wing disc harboring GstD1-GFP (green) and over-expressing *xrp1* (*UAS-xrp1*) in the posterior compartment (**A**) with quantification of GstD1-GFP signal intensity (**B**) (n = 8; two-sided Wilcoxon signed-rank test). (**C-D**) A wing disc of

the same genotype as in (**A**) immuno-stained for p62 (grey) (**C**) with quantification of p62 signal intensity (**D**) (n = 10; two-sided Wilcoxon signed-rank test). (**E-F**) A wing disc of the same genotype as in (**A**) immuno-stained for p-eIF2α (magenta) (**E**) with quantification of p-eIF2α signal intensity (**F**) (n = 8; two-sided Wilcoxon signed-rank test).

shown to induce proteotoxic stress, by causing accumulation of aggregogenic stress granules [38,39]. Therefore, we sought to induce high levels of p-eIF2α. Growth arrest and DNA-damage-inducible 34 (GADD34) is a Protein Phosphatase 1 (PP1) regulatory subunit, which causes p-eIF2α dephosphorylation by providing PP1 with target specificity for p-eIF2α [40]. As expected, *GADD34-RNAi* increased the levels of p-eIF2α (S5A and S5B Fig). *GADD34-RNAi* expression in the posterior compartment of wing discs also led to higher levels of p62 (Fig 4A and 4B) and of mono- and poly-ubiquitinated proteins (detected by the FK2 antibody; Fig 4C and 4D) than in the control anterior compartment. As these are both markers of protein aggregates [34,41], these data indicate that sustained eIF2α phosphorylation induces proteotoxic stress and protein aggregation. GADD34 knockdown also upregulated GstD1-GFP (Fig 4E and 4F) and p-JNK (Figs 4G and S5C). Thus, increased levels of p-eIF2α are sufficient to induce proteotoxic stress, the oxidative stress response, and JNK pathway activity, all of which are observed in *Rp/+* cells.

We then expressed *GADD34-RNAi* in a mosaic fashion to test whether it induces the loser status. *GADD34-RNAi* expressing cells were efficiently removed from wing discs in mosaic experiments (S5D Fig). Only a few fragments of cells remained, and these had been basally extruded from the epithelium (S5E Fig), consistent with competitive elimination. However, it was also possible that this was due to cell-autonomous activation of apoptosis. Thus, we designed an experimental strategy to obtain large *GADD34-RNAi* expressing patches of cells (S5F Fig) and directly compare the rate of apoptosis at the borders and centers of these patches, as increased border death is a hallmark of Minute cell competition [16,17,42]. We made use of Gal80^ts for conditional expression and placed larvae at the Gal80^ts permissive temperature after clone induction, to allow cells to expand without induction of transgene expression. We then activated *GADD34-RNAi* (and *GFP*) expression by moving larvae to the Gal80^ts restrictive temperature 24 hours before dissection (S5F Fig). This short period of *GADD34-RNAi* expression was sufficient to increase p-eIF2α (S5G Fig). Unlike control wild-type patches of cells, *GADD34-RNAi* expressing patches of cells had significantly higher levels of cell death at their borders than in the center, showing that they are subject to competitive elimination by wild-type cells (Figs 4H–4J and S6).

We next asked whether *GADD34-RNAi* induced cell competition depends on Xrp1. As Xrp1 and Irbp18 function upstream of proteotoxic stress in *RpS3/+* cells (Fig 1), we were surprised to find that co-expression of *xrp1-RNAi* with *GADD34-RNAi* resulted in a strong rescue of competitive elimination (Fig 5A–5C). Thus, elimination of *GADD34-RNAi* expressing cells is mediated by Xrp1, suggesting that Xrp1 can also function downstream of proteotoxic stress. Altogether, these data show that Xrp1 functions both upstream and downstream of proteotoxic stress, suggesting that a feed-forward loop between proteotoxic stress and Xrp1 exists in *Rp/+* cells.

Xrp1 knockdown also rescued the increased GstD1-GFP observed in *GADD34-RNAi* expressing compartments, bringing levels down to, or even slightly lower than, wild-type levels (Fig 5D–5F). Remarkably, Xrp1 knockdown was also able to partially rescue the increased p-eIF2α in *GADD34-RNAi* expressing compartments (Fig 5D, 5F and 5G), suggesting that removing Xrp1 breaks the feed-forward loop to proteotoxic stress, and therefore partially rescues the increased p-eIF2α levels in *GADD34-RNAi* expressing cells. Altogether these data suggest that Xrp1 is activated by proteotoxic stress. Consistently, we found that *GADD34-RNAi*

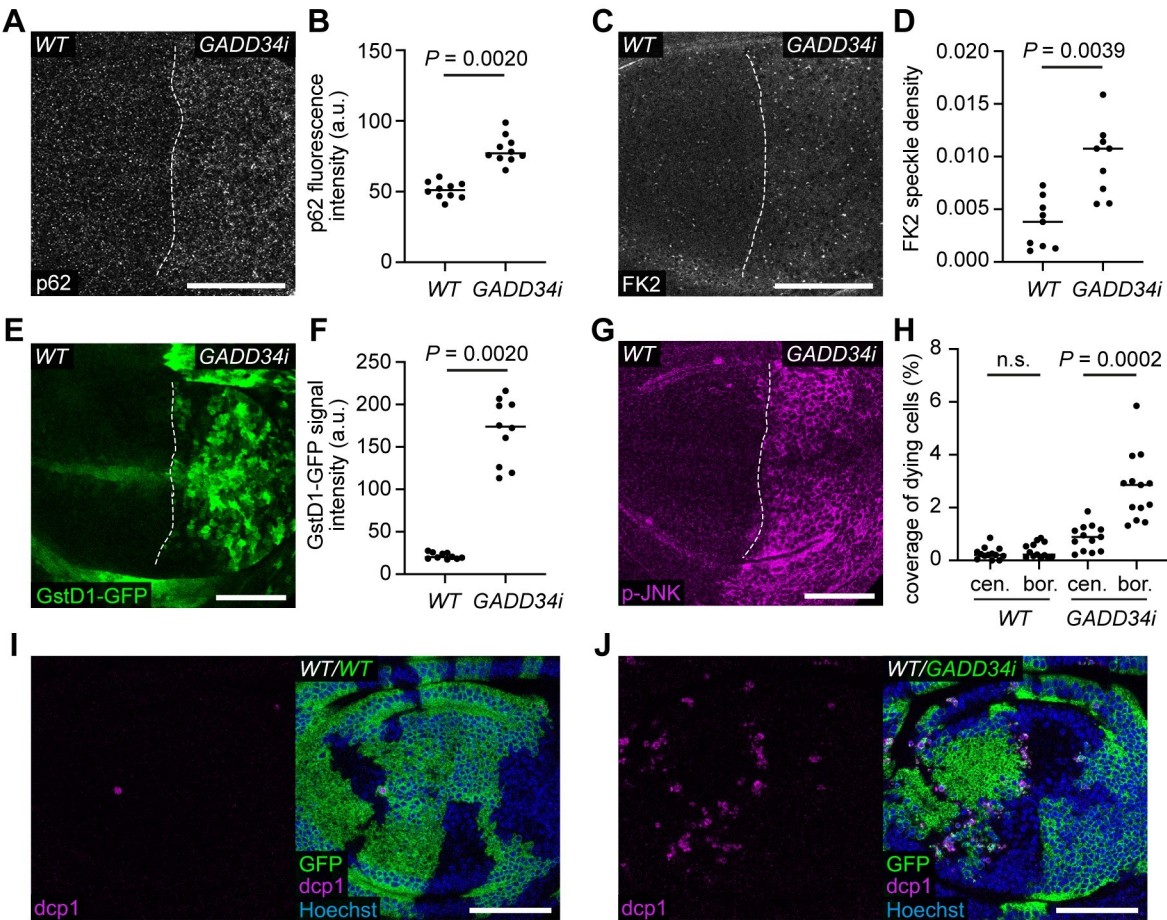

**Fig 4. GADD34 knockdown induces proteotoxic stress and the loser status.** (**A-B**) A wing disc carrying the GstD1-GFP reporter and expressing *GADD34-RNAi* (*GADD34i*) in the posterior compartment, immuno-stained for p62 (grey) (**A**) with quantification of p62 fluorescence intensity (**B**) (n = 10; two-sided Wilcoxon signed-rank test). (**C-D**) A wing disc of the same genotype as in (**A**), immuno-stained for FK2 (grey) to label mono- and poly-ubiquitinated proteins (**C**) with quantification of FK2 speckle density (**D**) (n = 9; two-sided Wilcoxon signed-rank test). (**E-F**) GstD1-GFP (green) in a wing disc of the same genotype as in (**A**), with quantification of GstD1-GFP signal intensity (**F**) (n = 10; two-sided Wilcoxon signed-rank test). (**G**) A wing disc of the same genotype as in (**A**), immuno-stained for p-JNK (magenta). (**H-J**) Wing discs harboring either GFP-positive WT cells (**I**) or GFP-positive *GADD34-RNAi* expressing cells (**J**) immuno-stained for dcp1 (magenta), with quantification of density of dying cells at the center (cen.) and border (bor.) of the GFP patches as indicated (**H**) (n = 13 and 13, respectively; two-sided Wilcoxon signed-rank test). The border defines cells within two cell diameters of the perimeter.

expressing compartments have significantly higher *xrp1-lacZ* signal than control compartments (Fig 5H and 5I).

How might Xrp1 be induced downstream of proteotoxic stress? During ER stress, the UPR induces eIF2α phosphorylation, which mediates global translation repression and selective translation of a subset of transcripts, including that of ATF4, which, in mammals, mediates expression of chaperones and proapoptotic genes, including *CHOP* [30,31]. Although no clear mammalian Xrp1 homolog exists, sequence homology and functional data suggest that Xrp1 may be functionally homologous to CHOP [25,29,43]. Indeed, overexpression of *ATF4* (also known as *cryptocephal* (*crc*) in *Drosophila*) was sufficient to upregulate both *xrp1* transcription and GstD1-GFP (S7A and S7B Fig). These data therefore suggest that Xrp1 can be transcriptionally activated by the UPR in *Drosophila*. We then tested whether ATF4 translation is increased in *RpS3/+* cells using the *crc-5'UTR-dsRed* translation reporter, which comprises the *dsRed* coding sequence

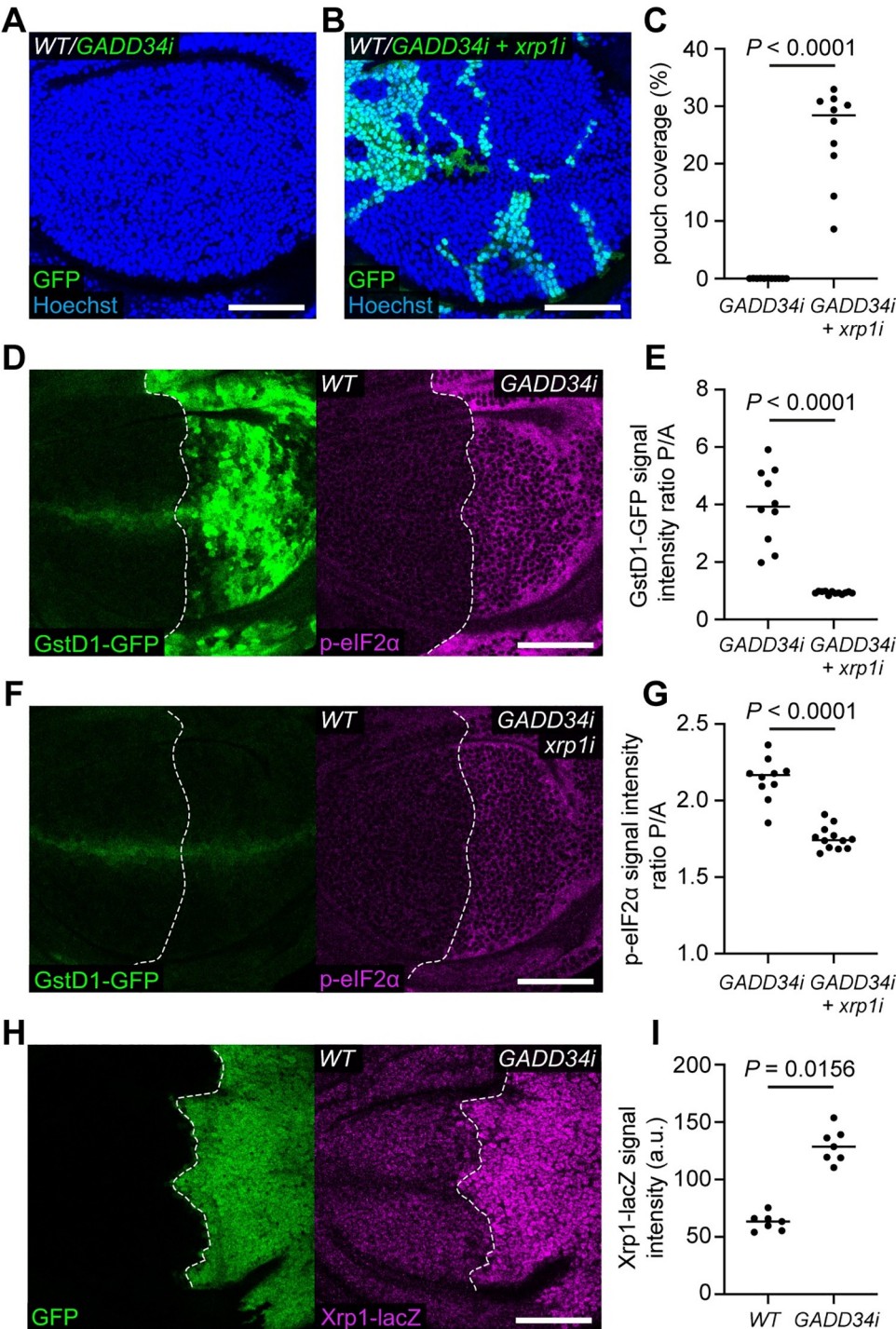

**Fig 5. A feed-forward loop between Xrp1 and proteotoxic stress.** (**A-C**) Wild-type wing discs harboring *GADD34-RNAi* (*GADD34i*) expressing cells (GFP positive) (**A**) or *GADD34-RNAi* and *xrp1-RNAi* (*xrp1i*) expressing cells (GFP positive) (**B**) with nuclei labelled in blue, and quantification of percentage coverage of the pouch (**C**) (n = 11 and 10, respectively; two-sided Mann–Whitney U-test). (**D-G**) Wing discs harboring GstD1-GFP (green) and expressing either *GADD34-RNAi* (**D**) or *GADD34-RNAi* and *xrp1-RNAi* (**F**) in the posterior compartment, immuno-stained for p-eIF2α (magenta), with quantification of the Posterior / Anterior (P/A) ratio of GstD1-GFP signal intensity (**E**) (n = 10 and 12, respectively; two-sided Mann–Whitney U-test) and the Posterior / Anterior (P/A) ratio of p-eIF2α signal intensity (**G**) (n = 10 and 12, respectively; two-sided Mann–Whitney U-test). (**H-I**) A wing disc carrying the *xrp1-lacZ* reporter and expressing *GADD34-RNAi* and *GFP* (green) in the posterior compartment,

immuno-stained with anti-β-galactosidase (magenta) (**H**), with quantification of *xrp1-lacZ* signal intensity (**I**) (n = 7; two-sided Wilcoxon signed-rank test).

placed downstream of the *ATF4 5'UTR* [44]. Activation of this reporter was clearly detected in *GADD34-RNAi* expressing cells (S7C and S7D Fig) but not in *RpS3/+* cells (S7E and S7F Fig), suggesting that ATF4 translation is not activated in *RpS3/+* cells. Furthermore, expressing a previously validated *ATF4-RNAi* line [45] did not rescue *xrp1* transcriptional upregulation in *RpS3/+* cells (S7G and S7H Fig). Therefore, this data suggests that, although ATF4 overexpression can activate Xrp1 transcription, ATF4 is not responsible for increasing *xrp1* expression in *Rp/+* cells.

Lastly, we investigated whether Xrp1 also plays a role in Nrf2-induced cell competition. We have previously shown that proteotoxic stress induces expression of the Nrf2 reporter GstD1-GFP [14] and that over-expression of *nrf2* is sufficient to turn otherwise wild-type cells into losers [24]. Consistently, *nrf2* expressing cells were readily eliminated from mosaic wing discs, with only a few tiny patches remaining at the time of dissection (Fig 6A and 6C). *xrp1-RNAi* significantly rescued the growth of *nrf2* expressing patches of cells (Fig 6B and 6C), indicating that Xrp1 functions downstream of Nrf2. Irbp18 knockdown also rescued *nrf2* expressing cells from elimination (Fig 6D–6F) confirming that Xrp1 functions, along with Irbp18, downstream of Nrf2. This suggests that in *Rp/+* tissues, Xrp1 is activated both by increased p-eIF2α, and by Nrf2.

## Discussion

We have provided evidence that a feed-forward loop between proteotoxic stress, Nrf2 and the Xrp1/Irbp18 complex is operational in *RpS3/+* cells (including in the absence of cell

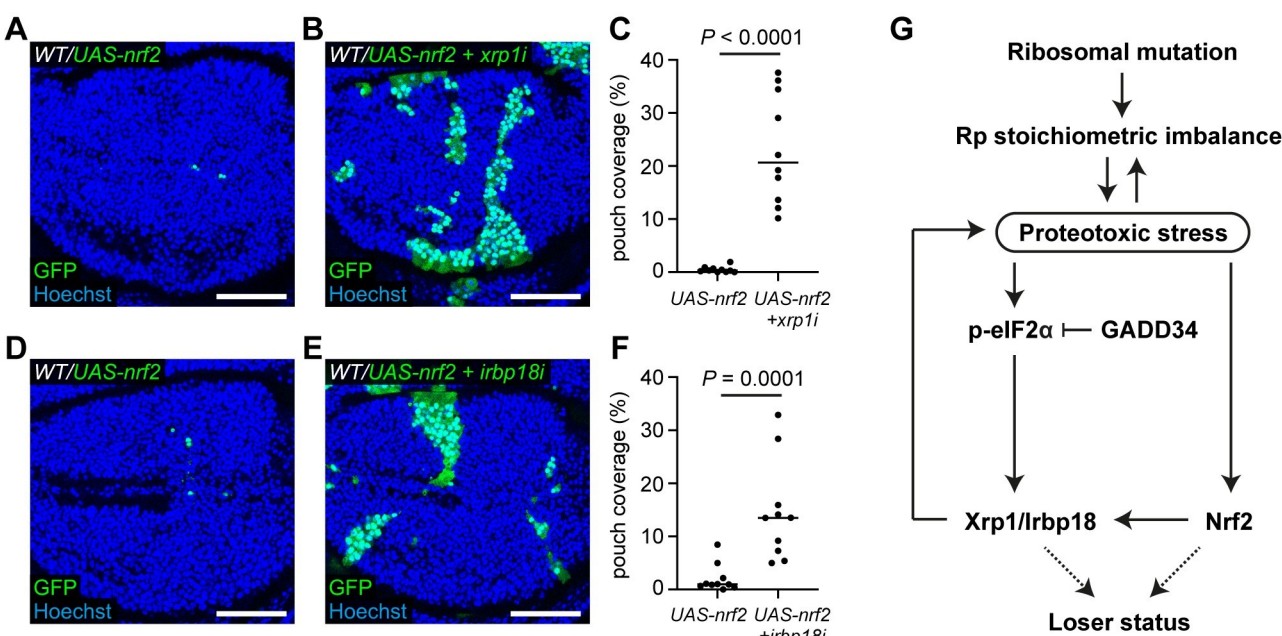

**Fig 6. Xrp1 and Irbp18 function downstream of Nrf2.** (**A-C**) Wild-type wing discs harboring *UAS-nrf2* expressing cells (GFP positive) (**A**) or *UAS-nrf2* and *xrp1-RNAi* (*xrp1i*) expressing cells (GFP positive) (**B**) with nuclei labelled in blue, and quantification of percentage coverage of the pouch (**C**) (n = 10 and 10, respectively; two-sided Mann–Whitney U-test). (**D-F**) Wild-type wing discs harboring *UAS-nrf2* expressing cells (GFP positive) (**D**) or *UAS-nrf2* and *irbp18-RNAi* (*irbp18i*) expressing cells (GFP positive) (**E**) with nuclei labelled in blue, and quantification of percentage coverage of the pouch (**F**) (n = 10 and 10, respectively; two-sided Mann–Whitney U-test). (**G**) Working model describing the role of the Xrp1/Irbp18 complex in *Rp/+* cells.

competition) and contributes to reducing their fitness during cell competition (Fig 6G). Our data suggests that an imbalance between SSU and LSU Ribosomal proteins generates an initial source of proteotoxic stress, independently of Xrp1. This leads to *xrp1* transcriptional upregulation, likely via p-eIF2α. As we have shown, Xrp1, together with Irbp18, generates further proteotoxic stress, in a feed-forward loop. This causes LSU ribosome proteins to accumulate, exacerbating the stoichiometric imbalance between LSU and SSU subunit components in *RpS3/+* cells. Knockdown of Xrp1 or Irbp18 rescues proteotoxic stress in *RpS3/+* cells, suggesting that this feed-forward loop is essential for build-up of proteotoxic stress and to reduce the competitiveness of *Rp/+* cells. We note that during the revision of this manuscript two other independent studies have reported relevant and complementary findings [46,47]. Nrf2 is also activated by proteotoxic stress and contributes to this feedback loop, either independently of p-eIF2α (as illustrated in Fig 6G), or downstream of p-eIF2α. Our data cannot distinguish between these two possibilities.

Our data indicate that *xrp1* upregulation is likely mediated by increased p-eIF2α levels. p-eIF2α accumulates in *Rp/+* cells [13,14], and increasing p-eIF2α in wild-type cells (by knocking down GADD34) leads to increased *xrp1* transcription (Fig 5), suggesting that p-eIF2α does, at least partially, contribute to *xrp1* transcription in *Rp/+* cells. p-eIF2α induces many transcriptional targets via stabilization of the transcription factor ATF4 [30,31]. This suggested that ATF4 may activate *xrp1*. Consistent with this, we found that ATF4 overexpression is sufficient to upregulate an *xrp1* transcriptional reporter in wing disc cells. However, we were surprised to find that *xrp1* upregulation does not seem to depend on ATF4 in *RpS3/+* cells. Indeed, ATF4 knockdown did not reduce *xrp1* transcription in *RpS3/+* cells. Furthermore, we were unable to detect stabilization of ATF4 in *RpS3/+* cells using a translational reporter (S7 Fig). These observations suggest that p-eIF2α upregulates *xrp1* transcription in *Rp/+* cells by an unknown, ATF4 independent, mechanism. Alternatively, the role of ATF4 may be masked by other inputs onto the *xrp1* promoter. For example, ATF4 knockdown could increase proteotoxic stress in *Rp/+* cells, by inhibiting the UPR, and this may upregulate other pathways that act on the *xrp1* promoter, thus masking any effect of ATF4 knockdown. This mechanism could involve Nrf2, since Nrf2 is also induced by proteotoxic stress [14] and since we have shown that Nrf2 induces cellular toxicity via *xrp1* (Fig 6). However, it is also possible that other factors activate Xrp1 in *Rp/+* cells.

Nrf2 plays a pro-survival role in many contexts, by activating a battery of genes that enable the metabolic adaptation to oxidative stress [23]. It is therefore counterintuitive that Nrf2 overexpression should induce the loser status and, at high expression levels, cell death [24]. Our work suggests that the toxicity of Nrf2 is at least in part due to Xrp1 function, as elimination of Nrf2 expressing cells is rescued by Xrp1 knockdown. Whether additional Nrf2 target genes contribute to the loser status remains to be established.

Besides Xrp1 or Irbp18 knockdown, the only other condition known thus far to rescue *xrp1* transcriptional upregulation in *Rp/+* cells is an *RpS12* point mutation, *RpS12^{97D}* [20,48]. However, the mechanism by which RpS12 affects *xrp1* transcription remains elusive. It will be important in future work to establish whether *RpS12* mutations rescue *xrp1* transcriptional activation upstream or downstream of proteotoxic stress.

Our results provide compelling evidence that Xrp1 and Irbp18 are responsible for inducing proteotoxic stress in *RpS3/+* cells. Firstly, knockdown of Xrp1 or Irbp18 rescues the accumulation of p62 labelled aggregates and rescues the increased p-eIF2α in *RpS3/+* cells (Fig 1). Secondly, overexpression of Xrp1 is sufficient to upregulate markers of proteotoxic stress in wild-type cells (Fig 3). Third, the presence of Xrp1 in *RpS3/+* cells worsens the imbalance of Ribosomal proteins, causing LSUs to accumulate (Fig 1). It will be crucial in future work to identify the relevant targets of Xrp1 that cause proteotoxic stress in *Rp/+* cells. Xrp1 may alter

expression of a single target, for example a gene encoding a component or regulator of the autophagy or proteasomal systems, which deregulates cellular proteostasis. Alternatively, several target genes may contribute to enhancing proteotoxic stress: if several subunits of multi-protein complexes are deregulated by increased Xrp1, this could lead to unassembled complexes, increasing the burden on the cellular degradation machinery in already stressed *Rp/+* cells. There may also be Xrp1 targets that contribute to the loser status without affecting proteotoxic stress. It is remarkable that, in addition to rescuing competitive elimination of *Rp/+* cells, loss of Xrp1 can rescue elimination of *mahj* deficient cells (Fig 2) and Nrf2 overexpressing cells (Fig 6). In *mahj* deficient cells, loss of Xrp1 was able to rescue the upregulation of p-eIF2α, suggesting that Xrp1 also promotes proteotoxic stress in *mahj* cells. It will be interesting to establish whether this is the case for Nrf2 expressing cells.

Xrp1 has been shown to play a role in a *Drosophila* model of Amyotrophic lateral sclerosis (ALS), a debilitating and lethal neurodegenerative disorder that can be caused by aggregogenic mutations in genes encoding RNA binding proteins, including TDP-43 and FUS, a member of the FET family of proteins [49]. TDP-43 and FUS also form cytoplasmic, ubiquitinated aggregates, in several other neurodegenerative disorders [50]. *Drosophila cabeza* (*caz*) is the single ortholog of the human FET proteins. Xrp1 is upregulated in *caz* mutants, and the pupal lethality, motor defects and dysregulated gene expression of *caz* mutants is rescued by *xrp1* heterozygosity [51]. Therefore, it is possible that the feed-forward loop we have uncovered is also active in this context: formation of cytoplasmic proteotoxic aggregates could stimulate *xrp1* expression, which could then induce further proteotoxic stress in a feed forward loop, resulting in neuronal toxicity. Understanding the relationship between Xrp1, proteotoxic stress and oxidative stress may thus be beneficial for the study of human proteinopathies.

## Methods

### Fly husbandry

Fly food composition is: 7.5g/L agar powder, 50g/L baker's yeast, 55g/L glucose, 35g/L wheat flour, 2.5% nipagin, 0.4% propionic acid and 1.0% penicillin/streptomycin. Eggs were collected for 24 hours in a 25˚C incubator and experimental crosses were then maintained in either an 18˚C incubator, a 25˚C incubator, or in a water bath set to a specific temperature. Mosaic wing discs were generated with the *hs-FLP* transgenic line by heat shocking crosses three days after egg laying in a 37˚C water bath. For experiments using temperature sensitive Gal80 (Gal80^ts) to control the timing and level of transgene expression, conditions were optimized for each experiment. All experimental conditions are listed in the Genotypes Table (Table 1). All *Drosophila* strains used are listed in the Key Resources Table (Table 2). Wing discs were dissected from wandering third instar larvae. For all experiments, egg collections, heat shocks, temperature shifts, dissections, and imaging were done in parallel for control and experimental crosses. All experiments comparing the effect of one UAS driven transgene to that of two UAS driven transgenes included an additional blank UAS, $40D^{UAS}$, to control for possible Gal4 titration effects, except for the experiment in S4B–S4D Fig, where $40D^{UAS}$ was not included due to the complexity of the genetic crosses. For mosaic competition experiments, all dissected larvae were of the same sex for both the control and experimental crosses. For half-half experiments, where the anterior compartment and posterior compartment were compared, sexes were not differentiated.

### Immunostaining

Wandering third instar larvae were dissected in phosphate buffered saline (PBS) and hemi-larvae were fixed in 4% paraformaldehyde for 20 minutes at room temperature. Tissues were

**Table 1. Genotypes Table.**

| Figure number/ panel | Genotype | Experimental conditions |
|---|---|---|
| | Main Figures | |
| 1A | GstD1-GFP/UAS-xrp1-RNAi; FRT82B, RpS3[Plac92], hh-Gal4/+ | 25˚C |
| 1D | GstD1-GFP/UAS-irbp18-RNAi; FRT82B, RpS3[Plac92], hh-Gal4/+ | 25˚C |
| 1G | GstD1-GFP/UAS-xrp1-RNAi; FRT82B, RpS3[Plac92], hh-Gal4/+ | 25˚C |
| 1I | GstD1-GFP/UAS-irbp18-RNAi; FRT82B, RpS3[Plac92], hh-Gal4/+ | 25˚C |
| 1L (control) | yw | 25˚C |
| 1L (RpS3$^{+/-}$) | FRT82B, RpS3[Plac92]/+ | 25˚C |
| 1L (RpS3$^{+/-}$, xrp1$^{+/-}$) | FRT82B, xrp1[m273], RpS3[Plac92]/+ | 25˚C |
| 2A | hs-FLP/+; 40D$^{UAS}$/+; act>CD2>Gal4, UAS-GFP/UAS-mahj-RNAi | 25˚C for 3 days, 20 min heat shock, 25˚C for 3 days |
| 2B | hs-FLP/+; UAS-xrp1-RNAi/+; act>CD2>Gal4, UAS-GFP/UAS-mahj-RNAi | 25˚C for 3 days, 20 min heat shock, 25˚C for 3 days |
| 2D | hs-FLP/+; 40D$^{UAS}$/+; act>CD2>Gal4, UAS-GFP/UAS-mahj-RNAi | 25˚C for 3 days, 25 min heat shock, 25˚C for 3 days |
| 2E | hs-FLP/+; UAS-xrp1-RNAi/+; act>CD2>Gal4, UAS-GFP/UAS-mahj-RNAi | 25˚C for 3 days, 25 min heat shock, 25˚C for 3 days |
| 3A | tub-Gal80[ts]/+; UAS-xrp1/en-Gal4, GstD1-GFP | 18˚C for 8–9 days, 29˚C for 24h |
| 3C | tub-Gal80[ts]/+; UAS-xrp1/en-Gal4, GstD1-GFP | 18˚C for 8–9 days, 29˚C for 24h |
| 3E | tub-Gal80[ts]/+; UAS-xrp1/en-Gal4, GstD1-GFP | 18˚C for 8–9 days, 29˚C for 24h |
| 4A | en-Gal4, GstD1-GFP/+; UAS-GADD34-RNAi/+ | 25˚C |
| 4C | en-Gal4, GstD1-GFP/+; UAS-GADD34-RNAi/+ | 25˚C |
| 4E | en-Gal4, GstD1-GFP/+; UAS-GADD34-RNAi/+ | 25˚C |
| 4G | en-Gal4, GstD1-GFP/+; UAS-GADD34-RNAi/+ | 25˚C |
| 4I | hs-FLP/+; tub>CD2>Gal4, UAS-CD8-GFP/+; tub-Gal80[ts] / + | 25˚C for 3 days, 35 min heat shock, 18˚C for 3 days, 29˚C for 24h |
| 4J | hs-FLP/+; tub>CD2>Gal4, UAS-CD8-GFP/+; tub-Gal80[ts] / UAS-GADD34-RNAi | 25˚C for 3 days, 35 min heat shock, 18˚C for 3 days, 29˚C for 24h |
| 5A | hs-FLP/+; 40D$^{UAS}$/+; act>CD2>Gal4, UAS-GFP/UAS-GADD34-RNAi | 25˚C for 3 days, 20 min heat shock, 25˚C for 3 days |
| 5B | hs-FLP/+; UAS-xrp1-RNAi/+; act>CD2>Gal4, UAS-GFP/UAS-GADD34-RNAi | 25˚C for 3 days, 20 min heat shock, 25˚C for 3 days |
| 5D | en-Gal4, GstD1-GFP/ 40D$^{UAS}$; UAS-GADD34-RNAi/+ | 25˚C |
| 5F | en-Gal4, GstD1-GFP/ UAS-xrp1-RNAi; UAS-GADD34-RNAi/+ | 25˚C |
| 5H | en-Gal4, UAS-GFP/+; FRT82B, xrp1-lacZ/UAS-GADD34-RNAi | 25˚C |
| 6A | hs-FLP/+; 40D$^{UAS}$/+; act>CD2>Gal4, UAS-GFP/UAS-nrf2 | 25˚C for 3 days, 20 min heat shock, 25˚C for 3 days |
| 6B | hs-FLP/+; UAS-xrp1-RNAi/+; act>CD2>Gal4, UAS-GFP/UAS-nrf2 | 25˚C for 3 days, 20 min heat shock, 25˚C for 3 days |
| 6D | hs-FLP/+; 40D$^{UAS}$/+; act>CD2>Gal4, UAS-GFP/UAS-nrf2 | 25˚C for 3 days, 20 min heat shock, 25˚C for 3 days |
| 6E | hs-FLP/+; UAS-irbp18-RNAi/+; act>CD2>Gal4, UAS-GFP/UAS-nrf2 | 25˚C for 3 days, 20 min heat shock, 25˚C for 3 days |
| | Supporting information Figures | |
| S1A | en-Gal4, UAS-GFP/ UAS-xrp1-RNAi; FRT82B, xrp1-lacZ/+ | 25˚C |
| S1C | en-Gal4, UAS-GFP/ UAS-irbp18-RNAi; FRT82B, xrp1-lacZ/+ | 25˚C |
| S1E | hs-FLP, UAS-CD8-GFP/+; VDRC[60100]/+; FRT82B, RpS3[Plac92], act>RpS3>Gal4/+ | 25˚C for 3 days, 25 min heat shock, 25˚C for 3 days |
| S1F | hs-FLP, UAS-CD8-GFP/+; UAS-xrp1-RNAi /+; FRT82B, RpS3[Plac92], act>RpS3>Gal4/+ | 25˚C for 3 days, 25 min heat shock, 25˚C for 3 days |
| S1H | hs-FLP, UAS-CD8-GFP/+; VDRC[60100]/+; FRT82B, RpS3[Plac92], act>RpS3>Gal4/+ | 25˚C for 3 days, 25 min heat shock, 25˚C for 3 days |
| S1I | hs-FLP, UAS-CD8-GFP/+; UAS-irbp18-RNAi /+; FRT82B, RpS3[Plac92], act>RpS3>Gal4/+ | 25˚C for 3 days, 25 min heat shock, 25˚C for 3 days |
| S2A | GstD1-GFP/40D$^{UAS}$; hh-Gal4/+ | 25˚C |
| S2D | GstD1-GFP/40D$^{UAS}$; FRT82B, RpS3[Plac92], hh-Gal4/+ | 25˚C |
| S3A | tub-Gal80[ts]/+; en-Gal4, GstD1-GFP/+; UAS-irbp18-HA/+ | 18˚C for 8–9 days, 29˚C for 24h |
| S3C | tub-Gal80[ts]/+; en-Gal4, GstD1-GFP/+; UAS-irbp18-HA/+ | 18˚C for 8–9 days, 29˚C for 24h |
| S3E | tub-Gal80[ts]/+; en-Gal4, UAS-GFP/+ | 18˚C for 8–9 days, 29˚C for 24h |

(*Continued*)

**Table 1.** (Continued)

| Figure number/ panel | Genotype | Experimental conditions |
|---|---|---|
| S3G | tub-Gal80[ts]/+; en-Gal4, UAS-GFP/+ | 18˚C for 8–9 days, 29˚C for 24h |
| S4A | tub-Gal80[ts]/+; en-Gal4, UAS-GFP, UAS-irbp18-RNAi/+; FRT82B, xrp1-lacZ/+ | 18˚C for 8–9 days, 29˚C for 24h |
| S4B | tub-Gal80[ts]/+; en-Gal4, UAS-GFP/UAS-xrp1 | 18˚C for 8–9 days, 29˚C for 24h |
| S4C | tub-Gal80[ts]/+; en-Gal4, UAS-GFP, UAS-irbp18-RNAi/UAS-xrp1 | 18˚C for 8–9 days, 29˚C for 24h |
| S5A | en-Gal4, GstD1-GFP/+; UAS-GADD34-RNAi/+ | 25˚C |
| S5D | hs-FLP/+; 40D$^{UAS}$/+; act>CD2>Gal4, UAS-GFP/UAS-GADD34-RNAi | 25˚C for 3 days, 20 min heat shock, 25˚C for 3 days |
| S5E | hs-FLP/+; 40D$^{UAS}$/+; act>CD2>Gal4, UAS-GFP/UAS-GADD34-RNAi | 25˚C for 3 days, 20 min heat shock, 25˚C for 3 days |
| S5G | hs-FLP/+; tub>CD2>Gal4, UAS-CD8-GFP/+; tub-Gal80[ts] / UAS-GADD34-RNAi | 25˚C for 3 days, 35 min heat shock, 18˚C for 3 days, 29˚C for 24h |
| S6A and S6B | hs-FLP/+; tub>CD2>Gal4, UAS-CD8-GFP/+; tub-Gal80[ts] / UAS-GADD34-RNAi | 25˚C for 3 days, 35 min heat shock, 18˚C for 3 days, 29˚C for 24h |
| S7A | tub-Gal80[ts]/+; en-Gal4, GstD1-GFP /+; FRT82B, xrp1-lacZ/UAS-ATF4 | 18˚C for 8–9 days, 29˚C for 24h |
| S7C | crc-5'UTR-dsRed/+; en-Gal4, GstD1-GFP/+; UAS-GADD34-RNAi/+ | 25˚C |
| S7E | crc-5'UTR-dsRed/+; en-Gal4, UAS-flp/+; FRT82B, RpS3[Plac92], ubi-GFP/FRT82B | 25˚C |
| S7G | en-Gal4, UAS-GFP/ATF4-RNAi; FRT82B, RpS3[Plac92], ubi-GFP/ FRT82B, xrp1-lacZ | 25˚C |

permeabilized with three 10-minute washes in PBST (0.25% triton in PBS) and blocked for 20 minutes in blocking buffer (4% fetal calf serum in PBST). Samples were incubated with primary antibodies diluted in blocking buffer at the concentration indicated in the Key Resources Table (Table 2) overnight at 4˚C. Samples were washed three times in PBST for 10 minutes and incubated with secondary antibodies and Hoechst diluted in blocking buffer at the concentration indicated in the Key Resources Table (Table 2) for 45-minutes at room temperature. After a further three 10-minute washes in PBST, wing discs were dissected from hemi-larvae and mounted in Vectashield (Vector laboratories) on borosilicate glass sides (no 1.5, VWR international).

## Proteomics

Sample preparation and Tandem Mass Tag (TMT) mass spectrometry were performed as described in [14].

## Image acquisition and processing

Images were acquired using a Leica SP8 confocal microscope with a 40x 1.3 NA P Apo Oil objective. Wing discs were imaged as z-stacks with each section corresponding to 1µm. Images were processed using Photoshop (Adobe Photoshop 2020) and Fiji (Version 2).

## Quantifications

Pouch coverage, cell death quantifications and fluorescence intensity quantifications were carried out using custom built Fiji scripts. All analysis focused on the pouch region of the wing disc. For measurements of pouch coverage, the percentage of the volume of the pouch occupied by GFP-positive cells was determined. For cell death quantifications the border is defined as any cell within a 2 cell-range of the boundary of the GFP-positive patch. Cell death measurements were normalized to the respective volume of the GFP-positive patch border or center, as measured in Fiji. For all scatter plots the horizontal line represents the median.

**Table 2. Key Resources Table.**

| | Antibodies | |
|---|---|---|
| Rabbit anti-p-eIF2α (1:500) | Cell signalling | Cat#3398T |
| Rabbit anti-Dcp1 (1:2000) | Cell signalling | Cat#9578S |
| Rabbit anti-Ref(2)P (1:5000) | Tor Erik Rusten [52] | N/A |
| Mouse anti-FK2 (1:1000) | Enzo Life Sciences | Cat#ENZ-ABS840-0100 |
| Rabbit anti-pJNK pTPpY (1:500) | Promega | Cat#V793B |
| Mouse anti-beta galactosidase (1:500) | Promega | Cat#Z3781 |
| Donkey anti-Rabbit IgG Alexa Fluor 555 (1:500) | Thermo scientific | Cat#A31572 |
| Donkey anti-Mouse IgG Alexa Fluor 555 (1:500) | Thermo scientific | Cat#A31570 |
| Hoechst 33342 solution (1:5000) | Thermo scientific | Cat#62249 |
| | *Drosophila* strains | |
| *Drosophila RpS3[Plac92]* | Bloomington | Cat#5627 |
| *Drosophila hh-Gal4/TM6b* | Jean-Paul Vincent | N/A |
| *Drosophila UAS-xrp1-RNAi* $^{KK104477}$ | VDRC | Cat#104477 |
| *Drosophila GstD1-GFP* | [32] | N/A |
| *Drosophila UAS-irbp18-RNAi* $^{KK110056}$ | VDRC | Cat#110056 |
| *Drosophila yw* | Daniel St. Johnston | N/A |
| *Drosophila FRT82B, xrp1[M273]* | Nicholas Baker | N/A |
| *Drosophila tub-Gal80$^{ts}$* | Jean-Paul Vincent | N/A |
| *Drosophila UAS-xrp1$^{long}$* | Shoichiro Kurata | N/A |
| *Drosophila en-Gal4* | Piddini lab stocks | N/A |
| *Drosophila UAS-GADD34-RNAi* | Bloomington | Cat#33011 |
| *Drosophila w+/w-; tub>CD2>Gal4, UAS-GFP; tub-Gal80$^{TS}$* | Bruce Edgar | N/A |
| *Drosophila en-Gal4, UAS-GFP* | Piddini lab stocks | N/A |
| *Drosophila FRT82B, xrp1$^{02515}$ (xrp1-lacZ)* | Nicholas Baker | N/A |
| *Drosophila UAS-ATF4-HA* | Bloomington | Cat#81655 |
| *Drosophila hs-FLP$^{122}$;; act>CD2>Gal4, UAS-GFP/TM6b* | Bruce Edgar | N/A |
| *Drosophila UAS$^{60101}$ (40D$^{UAS}$)* | VDRC | Cat#60101 |
| *Drosophila UAS-nrf2* | [32] | N/A |
| *Drosophila hs-FLP, UAS-CD8-GFP;; FRT82B, RpS3[Plac92], act>RpS3>Gal4/TM6b* | [14] | N/A |
| *Drosophila attP$^{60100}$ (empty attP)* | VDRC | Cat#60100 |
| *Drosophila UAS-mahj RNAi* | Bloomington | Cat#34912 |
| *Drosophila UAS-irbp18-HA* | FlyORF | Cat#F001677 |
| *Drosophila crc-5'UTR-dsRed* | [44] | N/A |
| *Drosophila ATF4-RNAi* | VDRC | Cat#2934 |

## Statistics and reproducibility

All data represented by the scatter plots including details of the specific statistical test used for each experiment are provided (S1 Data). Statistics were performed using GraphPad Prism (Prism 8). Univariate statistics were used to determine P-values. The statistical tests used were the Mann Whitney U-test for non-paired data, and the Wilcoxon matched-pairs signed rank test for paired data. P-value corrections for multiple comparisons were not considered due to the low number of comparisons. For experiments comparing across wing discs a minimum of three biological repeats were performed. For experiments with an internal control, a minimum of two biological repeats were performed. Experiments performed to validate reagents (e.g., testing efficacy of RNAi lines) were carried out at least once.

## Supporting information

**S1 Fig. Xrp1 or Irbp18 knockdown reduces *xrp1* transcription and rescues elimination of *RpS3*$^{+/-}$ cells.** (**A-B**) A wing disc carrying the *xrp1-lacZ* reporter and expressing *xrp1-RNAi* (*xrp1i*) and *GFP* (green) in the posterior compartment, immuno-stained with anti-β-galactosidase (magenta) and nuclei labelled with in blue (**A**), with quantification of *xrp1-lacZ* signal intensity (**B**) (n = 7; two-sided Wilcoxon signed-rank test). (**C-D**) A wing disc carrying the *xrp1-lacZ* reporter and expressing *irbp18-RNAi* (*irbp18i*) and *GFP* (green) in the posterior compartment, immuno-stained with anti-β-galactosidase (magenta) and nuclei labelled in blue (**C**), with quantification of *xrp1-lacZ* signal intensity (**D**) (n = 8; two-sided Wilcoxon signed-rank test). (**E-G**) Wild-type wing discs harboring *RpS3*$^{+/-}$ cells (GFP positive) (**E**) or *RpS3*$^{+/-}$ cells also expressing *xrp1-RNAi* (GFP positive) (**F**) with nuclei labelled in blue, and quantification of percentage coverage of the pouch (**G**) (n = 13 and 12, respectively; two-sided Mann–Whitney U-test). (**H-J**) Wild-type wing discs harboring *RpS3*$^{+/-}$ cells (GFP positive) (**H**) or *RpS3*$^{+/-}$ cells also expressing *irbp18-RNAi* (GFP positive) (**I**) with nuclei labelled in blue, and quantification of percentage coverage of the pouch (**J**) (n = 11 and 14, respectively; two-sided Mann–Whitney U-test).
(TIF)

**S2 Fig. *40D*$^{UAS}$ does not affect GstD1-GFP or p-eIF2α in wild type or *RpS3*$^{+/-}$ discs.** (A-C) A wild type wing disc carrying GstD1-GFP, a posterior Gal4 driver (*hh(hedgehog)-gal4*), and the *40D*$^{UAS}$ insertion used as a control for Gal4 titration. *40D*$^{UAS}$ did not markedly affect GstD1-GFP (green) or p-eIF2α (grey) (A). Cubitus interruptus (ci) (magenta) labels the anterior compartment. Quantification of GstD1-GFP (n = 9; two-sided Wilcoxon signed-rank test) and p-eIF2α (n = 9; two-sided Wilcoxon signed-rank test) signal intensity is shown in (B) and (C) respectively. (D-F) An *RpS3*$^{+/-}$ wing disc carrying GstD1-GFP, *hh-gal4*, and the *40D*$^{UAS}$ insertion. *40D*$^{UAS}$ did not affect GstD1-GFP (green) or p-eIF2α (grey) (D). Cubitus interruptus (ci) (magenta) labels the anterior compartment. Quantification of GstD1-GFP (n = 11; two-sided Wilcoxon signed-rank test) and p-eIF2α (n = 11; two-sided Wilcoxon signed-rank test) signal intensity is shown in (E) and (F) respectively.
(TIF)

**S3 Fig. Overexpression of Irbp18 or GFP does not induce proteotoxic stress.** (A-D) Wild-type wing discs over-expressing hemagglutinin (HA)-tagged Irbp18 (*UAS-irbp18-HA*) in the posterior compartment, immuno-stained for HA (green) and p-eIF2α (magenta) (A) or HA (green) and p62 (grey) (C) with quantification of p-eIF2α signal intensity (B) (n = 9; two-sided Wilcoxon signed-rank test) and p62 signal intensity (D) (n = 12; two-sided Wilcoxon signed-rank test). (E-H) Wild-type wing discs over-expressing GFP (green) in the posterior compartment and immuno-stained for p-eIF2α (magenta) (E) or p62 (grey) (G) with quantification of p-eIF2α signal intensity (F) (n = 12; two-sided Wilcoxon signed-rank test) and p62 signal intensity (H) (n = 7; two-sided Wilcoxon signed-rank test).
(TIF)

**S4 Fig. Irbp18 is partially required for Xrp1 to induce aggregates.** (**A**) *Xrp1-lacZ* expression in a wild-type wing disc that has been expressing *UAS-ibrp18-RNAi* in the posterior compartment (GFP positive) for 24h, as controlled with Gal80$^{ts}$, immuno-stained for anti-β-galactosidase (magenta). (**B-D**) Wild-type wing discs that have been expressing *UAS-xrp1* (**B**) or *UAS-xrp1* and *UAS-irbp18-RNAi* (**C**) in the posterior compartment (GFP positive) for 24h, immuno-stained for p62 (grey) with quantification of the posterior / anterior ratio of p62 signal intensity (**D**) (n = 10 and 11, respectively; two-sided Mann–Whitney U-test).
(TIF)

**S5 Fig. *GADD34-RNAi* cells induce JNK signalling and are eliminated from mosaic tissues.**
(**A-B**) A wing disc expressing *GADD34-RNAi* (*GADD34i*) in the posterior compartment and immuno-stained for p-eIF2α (magenta) (**A**) with quantification of p-eIF2α signal intensity (**B**) (n = 9; two-sided Wilcoxon signed-rank test). (**C**) Quantification of p-JNK signal intensity in wing discs expressing *GADD34-RNAi* in the posterior compartment (n = 10; two-sided Wilcoxon signed-rank test). (**D**) A wing disc harboring *GADD34-RNAi* expressing cells (GFP positive), generated in the absence of Gal80$^{ts}$, with nuclei labelled in blue. (**E**) A basal section of a wing disc harbouring *GADD34-RNAi* cells (GFP positive), generated in the absence of Gal80$^{ts}$, with nuclei labelled in blue, to show that only small, basally extruded patches of *GADD34-RNAi* expressing cells remain. Orthogonal views taken at the positions indicated by the yellow lines are shown to the right and bottom of the main image. (**F**) Schematic depicting experimental conditions for generating large *GADD34-RNAi* expressing patches of cells. (**G**) A wing disc with *GADD34-RNAi* expressing cells (GFP positive), generated with the experimental conditions depicted in (**F**), immuno-stained for p-eIF2α (magenta).
(TIF)

**S6 Fig. Examples of macro-output images for analysis of center and border death in *GADD34-RNAi* expressing cells (Fig 4H).** (**A-B**) Two examples of processed images for a single confocal section from wild-type wing discs harboring cells expressing *GADD34-RNAi* (GFP positive, top left panels) and immuno-stained for dcp1 (red, bottom left panels). GFP segmentation is shown (cyan, top right panels) and the center and border territories of the GFP patches are defined (top middle panels and bottom middle panels, with center territory indicated with orange lines and border territory indicated with green lines in the bottom middle panels). Segmentation of dcp1 positive cells overlayed with the center and border territories is shown in the bottom right panels: dcp1 positive regions in wild type cells are filled in yellow, border territory dcp1 positive regions are filled in red, and center territory dcp1 positive regions are filled in blue. Single confocal sections are shown here, but the analysis of center and border death was performed across multiple confocal sections, and the density of dying cells presented in Fig 4H is the percentage of the total volume of the border or center territory that is dcp1 positive.
(TIF)

**S7 Fig. ATF4 activation in *GADD34-RNAi* cells and *RpS3$^{+/-}$* cells.** (**A-B**) A wing disc carrying the *xrp1-lacZ* reporter and GstD1-GFP (green) and over-expressing *ATF4* (*UAS-ATF4*) in the posterior compartment, immuno-stained with anti-β-galactosidase (magenta) (**A**), with quantification of *xrp1-lacZ* signal intensity (**B**) (n = 11; two-sided Wilcoxon signed-rank test). (**C-D**) A wing disc carrying an ATF4 translation reporter (*crc-5'UTR-dsRed*) (grey) and GstD1-GFP (green) and expressing *GADD34-RNAi* in the posterior compartment (**C**) with quantification of *crc-5'UTR-dsRed* signal intensity (**D**) (n = 8; two-sided Wilcoxon signed-rank test). (**E-F**) A wing disc carrying *crc-5'UTR-dsRed* (grey) with an *RpS3$^{+/-}$* anterior compartment (GFP positive) and a wild-type posterior compartment (**E**) with quantification of *crc-5'UTR-dsRed* signal intensity (**F**) (n = 9; two-sided Wilcoxon signed-rank test). (**G-H**) An *RpS3$^{+/-}$* wing disc carrying *xrp1-lacZ* and expressing *ATF4-RNAi* (*ATF4i*) in the posterior compartment (GFP positive), immuno-stained with anti-β-galactosidase (magenta) (**G**), with quantification of the posterior / anterior (P/A) ratio of *xrp1-lacZ* signal intensity (**H**) (n = 8; one sample Wilcoxon signed-rank test).
(TIF)

**S1 Data. This source data file includes all raw data and analysis presented in Figs 1B, 1C, 1E, 1F, 1H, 1J, 1K, 1L, 2C, 2F, 3B, 3D, 3F, 4B, 4D, 4F, 4H, 5C, 5E, 5G, 5I, 6C, 6F, S1B, S1D, S1G, S1J, S2B, S2C, S2E, S2F, S3B, S3D, S3F, S3H, S4D, S5B, S5C, S7B, S7D, S7F and S7H.** (XLSX)

## Acknowledgments

We thank members of the Piddini lab for helpful discussions on the project. We thank the Wolfson Bioimaging Facility for access to microscopes and the University of Bristol Proteomics Facility for performing the TMT proteomic experiments and for bioinformatics support. We thank Flybase for providing a valuable resource. We thank Nicholas Baker, Hyung Don Ryoo, Shoichiro Kurata and Tor Erik Rusten for providing reagents.

## Author Contributions

**Conceptualization:** Paul F. Langton, Eugenia Piddini.

**Funding acquisition:** Eugenia Piddini.

**Investigation:** Paul F. Langton, Michael E. Baumgartner, Remi Logeay.

**Methodology:** Paul F. Langton.

**Project administration:** Eugenia Piddini.

**Supervision:** Eugenia Piddini.

**Writing – original draft:** Paul F. Langton.

**Writing – review & editing:** Paul F. Langton, Eugenia Piddini.

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
