## [Decision Letter · Decision Letter 0]

21 May 2021

Dear Dr Piddini,

Thank you very much for submitting your Research Article entitled 'Xrp1 and Irbp18 trigger a feed-forward loop of proteotoxic stress to induce the loser status' to PLOS Genetics.

The manuscript was fully evaluated at the editorial level and by independent peer reviewers. The reviewers appreciated the attention to an important problem, but raised some  concerns about the current manuscript. Based on the reviews, we will not be able to accept this version of the manuscript, but we would be willing to review a revised version.

If you decide to revise the manuscript for further consideration at PLOS Genetics, please aim to resubmit within the next 60 days, unless it will take extra time to address the concerns of the reviewers, in which case we would appreciate an expected resubmission date by email to plosgenetics@plos.org.

[LINK]

We are sorry that we cannot be more positive about your manuscript at this stage. Please do not hesitate to contact us if you have any concerns or questions.

Yours sincerely,

Norbert Perrimon

Associate Editor

PLOS Genetics

Gregory P. Copenhaver

Editor-in-Chief

PLOS Genetics

Reviewer's Responses to Questions

**Comments to the Authors:**

Reviewer #1: Summary

This manuscript connects the cell competition associated transcription factors Xrp1 and IRBP18 with proteotoxic stress. The authors show that both factors are needed for the induction of proteotoxic stress response in Minutes. Furthermore, their study reveals that Xrp1 is induced by Nrf2, which is essential for the loser state of Minutes, as well as proteotoxic stress. Combining the results, the authors suggest a feed-forward loop between Xrp1, proteotoxic stress and Nrf2, which ends in the “loser”state of Minutes.

The findings are original, building on previous work in the field and integrates Xrp1 into proteotoxic stress signaling. Consequently, the work will be of interest to many. The presented results largely support the paper’s conclusions however some additional work would be helpful.

Major points

- Figure 2 shows that ectopic expression of Xrp1 leads to induction of GstD1-GFP, p62 and peIF2α. Given the paper’s focus on both Xrp1 and IRBP18, the authors should also test if overexpression of IRBP18 leads to an increase in p62, p-eIF2α and GstD1. In addition, it would be important to exclude that overexpression per se induces proteotoxic stress (e.g. overexpress GFP or LacZ). The quantification suggests the induction of p-eIF2a and p62 is moderate. Overexpression of a neutral factor, as a control, would make the data more convincing. A minor aside related to this, the P values in Figure 2d and f are identical. Is this correct?

- Figure 4 shows that the induction of proteotoxic stress by the knockdown of GADD34 leads to an increase of peIF2α levels and Xrp1 expression. These results, in combination with the results from Figure 2, are interpreted as feed forward loop. The model illustrated in Figure 5g is predicted from earlier work on the UPR, summarized in reviews such as (Ron and Walter, 2007). For example, ATF4 being upstream of CHOP, a C/EBP protein proposed by Blanco et al 2020 to be the human Xrp1 homolog. A simple testable prediction of the models is the loss of ATF4 should decrease Xrp1 expression. To substantiate the claim, such an experiment should be done. The authors could perform either a knockdown of ATF4 in Minute flies to test if this leads to a reduction of Xrp1 expression. Minimally it should be tested if ATF4 expression is increased in Minutes. This would avoid the technical challenges such as exacerbation of cell competition (as discussed in paper)

- Given the connection in the paper between proteotoxic stress, cell competition and loser status, it would be great if the authors could show how Xrp1 expression is affected in a Minute situation when proteotoxic stress is alleviated. For example, does GADD34 overexpression block elimination.

Further comments

- In Figure 2b, d, and f the variance of the signal is quite high for the UAS-Xrp1 affected compartment. As noted above the variance and the p-values do not seem to match. To avoid confusion it would help to comment on this and to provide the raw data, including an example of how the quantification was done.

- The authors speak of RP +/- cells, when they only have used RpS3 +/- genotypes. This could be taken to imply that more RP genes were tested. If only RpS3 was tested this should be They should change this in the text to reduce misunderstandings.

- Figure 3- To faciliate repetition it should be made clearer how the quantification was made. Sufficient would be to provide a marked example – illustrating the clone border and center. In Figure 3j, it is not clear how many clones are present.

- Figure 4j vs 4k. it is difficult to reconcile the clear difference in signal seen in 4j with the moderate difference seen in 4k. Again by example it would be helpful clarify how the quantification was done.

- Figure 5g. GADD34 should be added to the proposed model, as it is one of the phosphatases regulating p-eIF2α.

- ER stress is transduced by several branches in the cell, do the authors predict that also other factors of ER stress play a role in regulating Xrp1 or forming the ‘loser’ state?

- ‘Feed-forward loop’ is a strong description for this mechanism, as it implements an endless cycle of increasing Xrp1 expression, which should lead to death -seen by several studies overexpressing Xrp1 via UAS/Gal4, but surprisingly Minute are viable. Could the authors comment on this?

- To give their claims more strength the author could also discuss the role of Xrp1 in the toxicity of the ALS-associated FUS orthologue caz mutant phenotype described by Mallik et al.,2018.

Reviewer #2: This paper adds nicely to an emerging body of work demonstrating that Rp gene haploinsufficiency leads to severe proteotoxic stress, and activates numerous stress signaling pathways including the ISR. The authors demonstrate mechanistic insight into how the Drosophila transcription factor Xrp1 mediates the intrinsic sensitivity of Rp+/- cells to elimination in mosaic tissues with competing wildtype cells. Several recent studies have identified Xrp1 as a key stress-induced gene that is required with its binding partner, the C/EBP homolog Irbp18, for the majority of phenotypes associated with ribosomal gene haploinsufficiency: the developmental delay, stress pathway activation (DNA damage, JNK, oxidative), cell competition associated gene expression (including expression of Xrp1 itself), and the competitive elimination of Rp+/- cells in mosaic tissues (Baillon 2018, Lee 2018, Ji 2019, Blanco 2020, Recasens-Alvarez 2021). Xrp1 is also induced in response to irradiation in a p53 dependent manner (Brodsky 2004, Akdemir 2007), mediates neurodegenerative toxicity (Gruenewald 2009, Mallik 2018), and is involved in transposon mobilization (Francis 2016), and is thus emerging as a major stress response factor that is induced by numerous different stresses.

Here, the authors follow up on their recent work (Kucinski et al 2017, Baumgartner et al 2021) with a careful set of experiments to investigate how Xrp1 contributes to the Rp+/- phenotype and the “loser status” of these cells in mosaic tissues. Their results lead them to propose that Xrp1 is induced by the transcription factor ATF4 in response to proteotoxic stress, and subsequently functions via a feed forward loop to regulate its own expression and numerous downstream effects that culminate in death of the cells. Overall the work is carefully done and clearly explained, and represents a significant contribution to our understanding of what makes some cells more susceptible to elimination from mosaic tissues.

Specific comments:

1. In their abstract (line 38) the authors conclude that “Xrp1 induce loser status by promoting proteotoxic stress”. But their data actually suggest a different conclusion: rather, that Xrp1 is induced in response to proteotoxic stress derived from an imbalance in ribosomal stoichiometry in Rp+/- cells, and then functions with Irbp18 to mediate the ISR. They show that loss of xrp1 reduced (“rescued”) p-eIF2a accumulation in Rp+/- cells and also reduced oxidative stress GSTD1-GFP; similar results with Irbp18 RNAi; p62 is also reduced by both. Together, the results suggest that much of the response to proteotoxic stress in Rp/+ cells is mediated by the Xrp1/Irbp18 complex (but is not inducing the stress itself).

2. The authors show that Xrp1 is required to eliminate Rp+/- cells in mosaics, as shown previously by other labs, and that cell elimination by competition between WT and mahj-RNAi expressing cells is also reduced by knockdown of Xrp1. They conclude that a common mechanism is at work in elimination of losers. This conclusion would be significantly stronger if they could show that loss of Xrp1 also prevented p-eIF2a in the mahj knockdown cells.

3. It is interesting that Irpb18 knockdown is not as efficient at preventing loser cell elimination as loss of Xrp1 (Fig. S1). Do the effects of overexpression of Xrp1 require Irbp18?

4. Based on their data that expression of ATF4 induces xrp1-lacZ and the literature, they suggest ATF4 is involved in initial Xrp1 upregulation. It would be nice to see this, however: If the authors block cell death in the Rp+/- cells (e.g., UAS-p35 or UAS-miRHG) would they be able to see that ATF4 is required for Xrp1 induction?

5. line 293: “this suggests that in Rp/+ tissues, proteotoxic stress activates Xrp1 by two routes, one via the UPR and ATF4, and the other via Nrf2.” I agree this makes sense, but this seems like a strong conclusion given that the data used to support it (Fig. 5) are from wildtype discs in which they express UAS-xrpr1-RNAi in UAS-Nrf2 overexpressing clones; could the conclusion be made clear that it is only one possibility?

6. Lines 297-299: “…proteotoxic stress and the Xrp1/Irbp18 complex, which is required for the elimination Rp+/- cells in competing mosaic tissues”. However, all of the markers of “loser status” used here (GstD1, p-JNK, Xrp1, p-eIF2a, p62, etc) are also required for the cell autonomous “Minute condition”; it should also be made clear here that these effects are intrinsic to the mutant, and are not restricted to cell competition in mosaics.

7. In the Fig. 5g model, where the Rp+/- condition/ Rp stoichiometric imbalance is proposed to lead to NRF2 induction, which then leads to Xrp1 induction. Interesting ideas, but speculation at this point – do they know that ATF4 is not required for Nrf2/GSTD induction? Also, the two dotted arrows to loser status make it more clear that this is speculative, but text seems to point to more linear relationships (i.e., Lines 293-294).

8. The authors propose that the initial proteotoxic stress comes from an imbalance between SSU and LSU ribosomal proteins (see note below, they make it seem here like it’s a new observation). Then, proteotoxic stress in Rp+/- cells induces Xrp1 expression. The Baker lab has provided evidence that RpS12 is specifically required for Xrp1 induction, which may be related to the SSU/LSU imbalance. Could the authors speculate about the role of RpS12 in the SSU/LSU imbalance and activation of Xrp1? The Baker lab has proposed that RpS12 triggers Xrp1 expression, perhaps these ideas are related. It would be appropriate to cite the previous work and speculate on links to their own interpretations.

9. Overall the authors have a tendency to over interpret and/or overstate their conclusions and sometimes forget to cite previous work showing the similar results. Examples: in lines 299-307: this paragraph describes their speculative model, but is written as if it was conclusive. Qualifying this description with “our data suggest a model in which…..” could mitigate this problem. Also, some of the statements made here are not exclusive to this current work, but seem implied to be: e.g. lines 299-300: this has also been shown by others (Lee and Baker, Baumgartner, Recasens-Alvarez, etc), so the authors should give them credit as well. The authors should also give more credit to Baillon et al 2018, who did chIP seq after over-expression of Xrp1 in wing discs and found that Xrp1 binds to its own locus and upregulates it, as well as many other genes associated with cell competition, including puc, hid and rpr. They also reported the homology with CEBP and postulated on its role.

10. Line 212: “If Xrp1 and Irbp18 are required in competition because they induce proteotoxic stress”, add “cell” in front of competition.

11. Cell death at clone borders vs centers: I always find this difficult to assess – in Fig. 3J, for example, the caspase stain appears to overlap with nuclei (in the same focal plane?) – whereas dying cells are primarily found to delaminate and fragment (as they show in FigS2e). If they do a 3D reconstruction of several sections (apical to basal), do they also see these delaminated caspase-marked cells? What is their distribution?

12. Fig. S2: The title to this figure is “GADD34 knockdown induces the loser status”. The authors find that Gadd34-RNAi induces p-eIF2a, pJNK and cell death, and go on to state that “increased border death is a hallmark of cell competition”, citing the Baker lab. Hallmark is a tricky word as it can imply an absolute. However, in other contexts of cell competition death is not primarily at clone borders, nor is it always at clone borders even in Rp+/- induced cell competition. To be more broadly accurate about “the loser status” (and perhaps more appropriately circumspect) I suggest that the authors change the title to something like “GADD34 knockdown phenocopies Rp+/- loser status”, or “GADD34 knockdown induces traits common to Rp+/- loser cells”.

13. On line 259, the authors state that titration of Gal4 by multiple UAS transgenes was controlled for in experiments using UAS-Gadd34-RNAi plus/minus UAS-xrp1-RNAi by adding “an inert” transgene to controls. I have two thoughts to consider here. First, this control RNAi from the VDRC does not appear lead to production of any RNA, making it formally possible that Pol II and associated factors are not recruited/assembled appropriately – is this really a good control for Gal4 titration, over say UAS-GFP or UAS-lacZ? Second, the authors indicate that UAS transgenes were controlled for in all subsequent experiments, which is very good (although of course should be standard practice when using multiple UASes). However, some experiments have even >2 UAS transgenes: e.g., Fig. 4b, 5a, d, e – were these controlled for as well?

14. The interesting statement is made that (line 309) “ATF4 plays a dual role: it promotes Xrp1 expression in Rp+/- cells, possibly along with Irbp18; however, it is also required for expression of chaperones..”. Is there any evidence from their RNA seq data (Kuscinski 2017) or other published transcriptome data (Baillon et al 2019, Boulan et al 2019, Baker lab) that suggest that Xrp1 is required for chaperone gene expression?

**Have all data underlying the figures and results presented in the manuscript been provided?**

Reviewer #1: None

Reviewer #2: Yes

PLOS authors have the option to publish the peer review history of their article (what does this mean?). If published, this will include your full peer review and any attached files.

Reviewer #1: No

Reviewer #2: **Yes: **Laura Johnston

---

## [Decision Letter · Decision Letter 1]

15 Nov 2021

Dear Eugenia,

We are pleased to inform you that your manuscript entitled "Xrp1 and Irbp18 trigger a feed-forward loop of proteotoxic stress to induce the loser status" has been editorially accepted for publication in PLOS Genetics. Congratulations! You may want to address a couple of minor suggestions from the reviewers - you can address those as you prepare your final draft for the production team (the editorial team will not need to re-evaluate).

Yours sincerely,

Norbert Perrimon

Associate Editor

PLOS Genetics

Gregory P. Copenhaver

Editor-in-Chief

PLOS Genetics

Comments from the reviewers (if applicable):

Reviewer's Responses to Questions

**Comments to the Authors:**

Reviewer #1: The additional experiments presented in the revised manuscript have improved the paper. New data has also led to a changed model. The authors could show that overexpression of IRBP18 alone does not induce the proteotoxic stress markers, p62 and pEIF2a, but it is required by the Xrp1 dependent induction of these markers.

A key revision to the model was the exclusion of ATF4. This was based on the following two experiments: i) using an ATF4 translational reporter they showed that ATF4 is not upregulated in RpS3(+/-) flies and ii) when ATF4 function was blocked in a RpS3(+/-) background Xrp1, the expression was not affected.

The new model is interesting and well supported. It could have been further fortified by testing if a reduction of pEIF2a in RpS3(+/-) decreases Xrp1 expression. Nevertheless, the authors satisfactorily addressed most of my previous concerns.

Reviewer #2: The authors addressed all of the Reviewers comments nicely, and I appreciate the hard work that the authors put into clarifying my questions and making many improvements to the paper. The experiments are well done and the data are important. While I am still not in complete agreement with the interpretation that Xrp1 actually induces ER stress and leads to cell death, as well as is induced as part of the ER stress response pathway activated by the Rp imbalance), their new data do provide some evidence that supports their view.

My only other suggestion is that the authors comment on (e.g. in their discussion) and cite Brown et al, eLife 2021, who showed that ER stress induces Xrp1 in a PERK dependent manner, and that the expression of GSTD1 and other antioxidant genes in ER stressed cells is very likely directly regulated by Xrp1.

**Have all data underlying the figures and results presented in the manuscript been provided?**

Reviewer #1: None

Reviewer #2: Yes

PLOS authors have the option to publish the peer review history of their article (what does this mean?). If published, this will include your full peer review and any attached files.

Reviewer #1: No

Reviewer #2: No

**Data Deposition**

http://datadryad.org/submit?journalID=pgenetics&manu=PGENETICS-D-21-00501R1

**Press Queries**

---

## [Editor Report · Acceptance letter]

1 Dec 2021

PGENETICS-D-21-00501R1 

Xrp1 and Irbp18 trigger a feed-forward loop of proteotoxic stress to induce the loser status 

Dear Dr Piddini, 

We are pleased to inform you that your manuscript entitled "Xrp1 and Irbp18 trigger a feed-forward loop of proteotoxic stress to induce the loser status" has been formally accepted for publication in PLOS Genetics! Your manuscript is now with our production department and you will be notified of the publication date in due course.

With kind regards,

Olena Szabo

PLOS Genetics

On behalf of:
